# Outcomes at Patient and Limb Levels in Peripheral Artery Disease by the Location of Atherosclerotic Lower Limb Lesions: An Observational Study from a High-Volume German Center

**DOI:** 10.3390/jcm14197037

**Published:** 2025-10-04

**Authors:** Anne Zimmermann, David J. F. Holstein, Paulina Stürzebecher, Paul Medicke, Annika Niezold, Maximilian Brunotte, Samira Zeynalova, Armin Wiegering, Daniel Seehofer, Andrej Schmidt, Sabine Steiner, Dierk Scheinert, Daniela Branzan, Konstantin Uttinger

**Affiliations:** 1Department of Visceral, Transplant, Thoracic and Vascular Surgery, Leipzig University Hospital, 04103 Leipzig, Germany; anne.zimmermann@medizin.uni-leipzig.de (A.Z.); a.niezold@gmail.com (A.N.); daniel.seehofer@medizin.uni-leipzig.de (D.S.); 2Department of Vascular Surgery, Diakonissenkrankenhaus Leipzig, 04177 Leipzig, Germany; 3Department of Cardiology, Leipzig University Hospital, 04103 Leipzig, Germany; 4Department of Vascular and Endovascular Surgery, Klinikum Rechts der Isar, Technical University of Munich, Ismaninger Str. 22, 81675 Munich, Germany; 5Institute for Medical Informatics, Statistics and Epidemiology, Leipzig University, 04107 Leipzig, Germany; 6Leipzig Research Centre for Civilization Diseases, Leipzig University, 04103 Leipzig, Germany; 7Department of General, Visceral, Transplant and Thoracic Surgery, Frankfurt University Hospital, Goethe University Frankfurt/Main, Theodor-Stern-Kai 7, 60596 Frankfurt/Main, Germany; 8Division of Angiology, Department of Internal Medicine, Neurology and Dermatology, University Hospital Leipzig, 04103 Leipzig, Germany; andrej.schmidt@medizin.uni-leipzig.de (A.S.); sabine.m.steiner@meduniwien.ac.at (S.S.);; 9Helmholtz Institute for Metabolic, Obesity and Vascular Research (HI-MAG), Helmholtz Zentrum Munich, University of Leipzig and University Hospital Leipzig, 04103 Leipzig, Germany; 10Department of Angiology, Universitätsklinik für Innere Medizin II, University Hospital Wien, Währinger Gürtel 18-20, 1090 Wien, Austria; 11Frankfurt Cancer Institute, Georg-Speyer-Haus, Paul-Ehrlich-Str. 42-44, 60596 Frankfurt/Main, Germany

**Keywords:** peripheral artery disease, amputation-free survival, major amputation, atherosclerotic lesion, location of atherosclerotic lesions, atherosclerotic lesion

## Abstract

**Background:** In Peripheral Artery Disease (PAD), there is an association between risk factors, the location of atherosclerotic lesions, and outcomes. **Methods:** This is a retrospective single-center analysis of adult PAD patients admitted between 2018 and 2021 with a follow-up until the end of 2023. Lesions were allocated to “suprainguinal”, “infrainguinal-to-popliteal”, “infrapopliteal”, “two of three levels” and “all three levels” categories based on angiogram findings. The primary endpoint at the patient level was amputation-free survival and was major adverse limb events (MALEs) at the limb level. **Results:** A total of 2067 patients with 2633 affected limbs were analyzed, and 28.8% were female. At first admission, the median age was 68, and the most frequent PAD Fontaine stage was IIb (44.9%). Lesions were suprainguinal in 11.6%, infrainguinal-to-popliteal in 18.3%, infrapopliteal in 11.4%, two levels in 36.0%, and all three levels in 8.3%. Over 1020 days as the median follow-up, amputation-free survival was 67.6%, highest (92.5%) for suprainguinal lesions, and lowest (59.3%) for infrapopliteal lesions. At the patient level, the risk of major amputation or death was highest in infrapopliteal lesions and was equally likely in cases of two or three affected locations and was reduced in infrainguinal-to-popliteal lesions (Hazard Ratio, HR 0.62, 95% CI 0.44–0.87, *p* = 0.007) and suprainguinal lesions (HR 0.42, 95% CI 0.21–0.79, *p* = 0.008). At the limb level, compared to lesions in all three locations, the risk of MALEs was reduced in infrainguinal-to-popliteal lesions (HR 0.51, 95% CI 0.27–0.98, *p* = 0.044) and was equally likely in all other cases. **Conclusions:** Amputation-free survival was lowest in cases of infrapopliteal lesions or multi-level disease. At the limb level, isolated infrainguinal-to-popliteal lesions were associated with the lowest risk of MALEs.

## 1. Introduction

According to the American Heart Association, cardiovascular disease is currently the leading cause of death globally [1], comprising coronary heart disease, stroke, and Peripheral Artery Disease (PAD), which affects an estimated 113 million people worldwide [2]. Due to high amputation rates and restrictions in mobility, PAD has a negative impact on quality of life [3,4] and is associated with generalized atherosclerosis and a high mortality following cardiovascular events [1,5,6,7].

Besides the PAD stage, patient symptom severity, age, and comorbidity, the atherosclerotic lesion location is decisive in determining adequate invasive therapeutic approaches [8,9,10,11], i.e., interventional, surgical, or combined techniques. Currently, only symptomatic lesions are subject to interventional or surgical treatment, while any type of atherosclerotic lesion entails preventive conservative and medical treatment.

In manifest PAD of the lower extremities, a high inter-individual variability of segmental atherosclerotic distribution patterns is observed, with atherosclerotic lesions categorized as suprainguinal, infrainguinal-to-popliteal, and infrapopliteal. It has been described that atherosclerotic lesions in different locations can have distinct properties [12,13], angiographic particularities [10], and coincident risk factors [14]. In addition, an association between the location of atherosclerotic lesions in PAD and mortality risk was found [15]. Beyond this association, it has since been demonstrated that an extended lower extremity atherosclerotic lesion burden translates to an increased risk of cardiovascular events in PAD patients [16].

To date, while these insights into the association between the atherosclerotic disease location, burden, and outcome have been described, there is limited evidence parallelling this association at the patient level, i.e., mortality risk, and the limb level, i.e., risk of limb events, in one cohort. This view might enable intensified preventive strategies for limb salvage programs vs. mortality reduction in PAD subgroups at an elevated risk.

To address this, it was the aim of this study to investigate a longitudinal association between patterns of atherosclerotic lesions in PAD patients and outcome parameters at patient and limb levels.

## 2. Methods

### 2.1. Study Design and Data Acquisition

This is a retrospective single-center analysis of patients with Peripheral Artery Disease (PAD) as main ICD diagnosis (ICD-10 GM I70.2) admitted to the University Hospital Leipzig between 1 January 2018 and 31 December 2021. Only adult patient records were analyzed. All re-admissions for PAD to the University Hospital Leipzig until 31 December 2021 were included for further analysis and for amputation status. Revascularization procedures in this analysis were performed by the vascular surgery department or the angiology department.

To record deaths among the analyzed patients during the observation period after discharge, a query was made to the Saxon Registration Register (“Sächsisches Melderegister”) for all patients, and life status was obtained on 30 November 2023 for all patients.

The ethics board of the University Leipzig authorized this analysis (vote 304/22-ek); no patient consent was required. The work has been reported in line with the STROCSS [17,18] criteria and the STROBE guidelines [19]. This study meets all five of the CODE-EHR minimum framework standards for the use of structured healthcare data in clinical research, with zero out of five standards meeting preferred criteria [20]. It was registered with a Research Registry UIN (researchregistry11116) (https://researchregistry.knack.com/research-registry#home/registrationdetails/67e002765a6bd2033d7aa789/, accessed on 3 October 2025). 

If patients were admitted or re-admitted due to any different main diagnosis than PAD, these records were not included in the analysis, introducing possible bias toward lower revascularization procedures while maintaining specificity in PAD patient selection. To avoid bias, patients with a known previous amputation surgery in their secondary diagnoses were excluded for further analysis at limb level. Limbs with no angiogram or revascularization during this period were not included in this analysis.

Procedures were coded using procedural codes (OPS codes, “Operationen und Prozedurenschlüssel”). Surgical revascularization was defined as endarterectomy and bypass surgery. Endovascular revascularization was defined as interventional atherectomy, thrombectomy, thrombolysis, balloon angioplasty, and/or usage of any kind of stenting device. Performance of any amputation above the ankle or a surgical limb re-revascularization after primary revascularization was defined as major adverse limb event. An amputation below the ankle or revision surgery, which was identified using procedural code for revision/debridement surgery, were defined as minor limb event (procedure codes in Appendix A).

### 2.2. Definition of Primary and Secondary Endpoints

At patient level, the primary endpoint was amputation-free survival (survival without above the ankle amputations), and overall survival was the secondary endpoint at patient level. The primary endpoint at limb level was major adverse limb events (MALEs) during the time under observation. The secondary endpoint at limb level was minor limb events during the time under observation.

Time under observation was defined as the time between the first admission date and the last discharge date for endpoints at limb level and as time between the first admission date and 30 November 2023 for endpoints at patient level.

### 2.3. The Allocation of the Level of Atherosclerotic Lesions and the Definition of Primary Revascularization Procedures

An atherosclerotic lesion seen on angiogram was considered relevant if it was explicitly mentioned as stenosis, referred to as occluded and revascularizable, or led to revascularization in the same session. In the case of an infrapopliteal location, the mere expression of one vascular or two vascular supplies was not considered a lesion. The first allocation in time was chosen for further analysis and is referred to as the primary level of atherosclerotic lesions (suprainguinal, infrainguinal-to-popliteal, infrapopliteal, at least two of the above, all three); if this first angiogram did not contain information on one or more levels and if in a following angiogram there was an atherosclerotic lesion found at that level, the primary allocation was updated to include that information. For a subset of written findings of angiograms, no side allocation was available.

The level of atherosclerotic lesions refers to the respective limb for limb-level analyses, whereas in the case of analyses at patient level, the stated level of atherosclerotic lesions was present in either of the two limbs. Diagnoses were coded without side coding in the available data.

For the identification of the location of the level of atherosclerotic lesions (suprainguinal, infrainguinal-to-popliteal, infrapopliteal) at limb level, all written findings of all angiograms during all admissions were manually reviewed and assigned to the respective location at limb level by blinded trained medical personnel. Lesions in the external iliac artery, internal iliac artery, and/or common iliac artery are referred to as suprainguinal; lesions in the common femoral artery, femoral artery, and/or deep femoral artery are referred to as infrainguinal-to-popliteal. From the P1-segment of the popliteal artery onward, lesions are referred to as infrapopliteal. The first revascularization procedure per limb during the time under observation was defined as primary revascularization. In the case of the primary revascularization being “interventional and surgical”, the two procedures may or may not have occurred simultaneously but were performed during the same admission. If angiograms were conducted externally, their written findings could not be accessed. No ultrasound, CT angiogram, or MR angiogram results were accessed.

### 2.4. Statistical Analysis

For risk stratification of comorbidity, a three-category frailty score based on secondary diagnoses was determined for each patient (low, intermediate, and high risk; Frailty I to III, respectively) [21,22]. Medication details were obtained based on noted co-medication for all patients with interventions; no information on compliance was available.

For primary and secondary endpoints during the time under observation, Kaplan–Meier curves were used for depiction as well as adjusted survival curves. To identify covariates, the “disjunctive cause criterion” based on clinical assumptions was deployed to control for available pre-exposure covariates (exposure being hospital admission and revascularization procedures) that are a cause of the exposure (age, diabetes mellitus, and frailty), the outcome (age, diabetes mellitus, frailty, sex, primary revascularization, PAD stage at first admission), or both [23,24]. Due to unknown interrelation, no causal acyclic diagram was available. We excluded significant multicollinearities among confounding variables. To adjust for covariates, a Cox proportional hazards model was deployed using the Efron method for ties [25]. The proportional hazards assumption was tested on the basis of Schoenfeld residuals; interactions between covariates were accounted for using interaction terms. A multivariable fractional polynomial model was built from the initial set of predictors for the Cox regression model to assess the linearity assumption using the backfitting model selection algorithm [26]. A nominal *p* value of 0.05 was set for variable and fractional polynomial selection for non-binary variables, except for primary revascularization and location of the atherosclerotic lesion, which were forced into the model. Likelihood ratio tests were used to assess the accuracy of the Cox regression model. Log-rank tests were deployed to assess discrimination of unadjusted Hazard Ratios. For the primary location of the level of atherosclerotic lesions, the level with the highest fraction of the defined endpoint was chosen as reference in the Cox regression model. On patient level, post hoc analyses stratified by claudication (Fontaine IIa and IIB) vs. CLTI—i.e., chronic limb-threatening ischemia, with Fontaine III and IV as surrogates—were performed.

Quantitative data are stated as median and interquartile range. Where appropriate, 95% confidence intervals (95% CIs) were computed.

All calculations were performed with Stata 16.1 (StataCorp LP, College Station, TX, USA).

## 3. Results

### 3.1. Study Population

After the application of the inclusion and exclusion criteria, 2067 individual patients admitted to the University Hospital Leipzig for Peripheral Artery Disease (PAD) between 1 January 2018 and 31 December 2021 were included in the present study (Appendix A, details in Appendix A). At the limb level, due to a coded previous amputation surgery (ICD Z98.4–Z98.9) in 183 admissions and a total of 1285 asymptomatic limbs (no performed angiogram or revascularization), a total of 2633 lower limbs were analyzed (Appendix A). The median age at first admission was 68 (interquartile range, IQR, 61–78), and 595 patients were female (28.8%). PAD Fontaine stages at first admission were coded as IIb in 44.9%, as IV in 35.4%, as III in 13.1%, and as unknown or coded as IIa in 6.9%. Co-medication included any antiplatelet therapy in 54.6% and statin therapy in 70.8%. The median body mass index was 26 kg/m^2^ (IQR 23–30) (Table 1). PAD Fontaine stages differed by the lesion allocation at the limb level: in cases of isolated infrapopliteal lesions, Fontaine III or IV was present in 72.4% vs. 15.1% in cases of isolated suprainguinal lesions. Fontaine IIa or IIb were the most common (85.0%) in cases of isolated suprainguinal lesions, while it was present in 27.5% in infrapopliteal lesions (Appendix A).

On the basis of defined secondary diagnoses [21,22], the patient frailty was 1 (low frailty) in 73.4%, 2 (medium frailty) in 21.0%, and 3 (high frailty) in 5.6% (Table 1). Coronary heart disease was coded in 36.4% (Table 1, Appendix A). In 45.5%, diabetes mellitus was coded as a secondary diagnosis. At least one re-admission occurred in 31.9%, with a median inter-admission interval of 63 days (IQR 34–156 days) (Table 1). At the patient level, in 12.1% no lesion allocation was possible due to missing angiograms. In 12.5%, an isolated infrapopliteal lesion was found, whereas in 13.3% and 6.4% an isolated primary lesion was found in an infrainguinal-to-popliteal location and in a suprainguinal location, respectively. In 42.1%, two levels were affected, and in 13.6% lesions in all three levels were present (Table 2). At the patient level, the overall number of primary revascularization procedures was 565 (Table 2). At the limb level, in 380 limbs (14.4%) no allocation was possible due to missing angiograms. In 7.6%, an isolated infrapopliteal lesion was found, whereas in 12.3% and 7.8% an isolated primary lesion was found in an infrainguinal-to-popliteal and in a suprainguinal location, respectively. In 36.0%, two levels were affected, and in 8.3% lesions in all three levels were present (Table 3, patient characteristics by primary lesion in Appendix A). At first admission, patients were oldest in cases of primary infrapopliteal lesions (75 years in median, IQR 66–82) and youngest in cases of primary suprainguinal lesions (59 years in median, IQR 55–65; overall *p* < 0.001). The fraction of patients classified as low frailty was highest in cases of suprainguinal lesions (90.2%) and lowest in cases of infrapopliteal lesions (62.4%, overall *p* < 0.001).

### 3.2. Primary and Secondary Endpoints at Patient Level

In-hospital death occurred in 4.7%, and was lowest in isolated infrainguinal-to-popliteal lesions (0.7%). Death during the follow-up after the last hospital admission was highest in isolated infrapopliteal lesions (35.0%). Death during follow-up was increased after an in-hospital occurrence of major and minor limb events (37.4% and 42.9% vs. overall 27.2%, respectively) (Appendix A).

At the patient level, amputation-free survival was 67.6% over a median observation time of 1020 days (IQR 658–1412). The median time to the above-ankle amputation or death was, in cases of occurrence, 340 days (IQR 58–843) (Table 2).

Amputation-free survival differed by primary lesion; it was 92.5% in cases of an isolated suprainguinal lesion, 59.3% in isolated infrapopliteal lesions, and 62.9% in lesions in all three locations (Table 2, Figure 1A).

At the patient level, no details, i.e., time coding and/or side coding, of the coded revascularization procedure were available in 4.4% of patients.

Stratified by primary revascularization, which was no intervention in 72.7%, interventional revascularization in 12.2%, surgical revascularization in 10.4%, and both interventional and surgical revascularization in 4.7% (Table 2), the amputation-free survival ranged between 72.7% and 71.9% (no intervention and interventional revascularization, respectively) and 56.7% (both interventional and surgical revascularization) (primary endpoint by primary revascularization and by primary lesion at patient level in Appendix A).

The overall survival as a secondary endpoint was 69.4%, and the median time to death in cases of occurrence was 407 days (IQR 88–899). The overall survival ranged between 92.5% (isolated suprainguinal lesion) and 61.2% (isolated infrapopliteal lesion) by the primary lesion (Table 2, Figure 2A; secondary endpoint by primary revascularization and by primary lesion at patient level in Appendix A).

In a Cox regression model, major amputation or death significantly increased with age (Hazard Ratio, HR, 1.05 per year, 95% confidence interval, CI 1.04–1.06), was more likely in males (HR 1.25, 95% CI 1.03–1.51), increased with frailty (frailty 3 HR 1.02, 95% CI 1.00–1.04), was less likely in claudication (Fontaine IIa and IIB; CLTI, i.e., chronic limb-threatening ischemia, Fontaine III and IV as surrogate as reference) (HR 0.99, 9.98–0.99), and increased in cases of diabetes mellitus (HR 1.02, 95% CI 1.01–1.02). In reference to no primary revascularization, the primary endpoint was less likely in cases of primary interventional revascularization (HR 0.72, 95% CI 0.55–0.94), equally likely in cases of primary surgical revascularization (HR 1.32, 95% CI 0.95–1.83), and more likely in cases of a primary interventional and surgical revascularization (HR 2.24, 95% CI 1.54–3.26). In reference to the primary lesion being infrapopliteal, the occurrence of a major amputation or death was less likely in infrainguinal-to-popliteal (HR 0.62, 95% CI 0.44–0.87) and suprainguinal (HR 0.42, 95% CI 0.21–0.79) lesions and was equally likely in cases of two of three lesions (HR 0.99, 95% CI 0.79–1.25) and in cases of lesions in all three locations (HR 1.19, 95% CI 0.88–1.59) (Figure 1A, Figure 3A, results in detail in Appendix A).

In the multivariable Cox regression, death was associated with the same factors as major amputation or death, i.e., it significantly increased with age, was more likely in males, increased with frailty and in the presence of CLTI, and was more likely in diabetics. It was, in reference to no primary revascularization, less likely in cases of primary interventional revascularization, equally likely in cases of primary surgical revascularization, and more likely in cases of a primary interventional and surgical revascularization. In reference to isolated infrapopliteal lesions, death was less likely in cases of suprainguinal or infrainguinal-to-popliteal lesions (HR 0.49, 95% CI 0.25–0.96, and HR 0.69, 95% CI 0.48–0.98, respectively) and equally likely in cases of two or three of three locations (Figure 2A, Appendix A). As a post hoc stratified analysis, in claudicant patients, the risk of major amputation or death was reduced in cases of suprainguinal lesions (HR 0.30, 95% CI 0.10–0.96) and was equally likely in all other lesion locations, while there was no association between the lesion location and death in this subgroup. In CLTI patients, the risk of major amputation or death was increased in cases of lesions in all three locations (HR 1.40, 95% CI 1.00–1.94) and was equally likely in all other lesion locations, with no association between the lesion location and death (Appendix A).

### 3.3. Primary and Secondary Endpoints at Limb Level

At the limb level, the occurrence of major adverse limb events was 6.2% overall. The median time to major adverse limb events, in cases of occurrence, was 122.5 days (IQR 39.5–354.5) (Table 3).

The occurrence of major adverse limb events differed by primary lesion; i.e., it was lowest in cases of isolated suprainguinal lesions (2.6%) and was highest (12.8%) in cases of lesions in three of three locations (Table 3, Figure 1B).

Stratified by primary revascularization at the limb level, which was no intervention in 83.1%, interventional revascularization in 4.8%, surgical revascularization in 8.9%, and both interventional and surgical revascularization in 3.2% (Table 3), the occurrence of major adverse limb events ranged between 2.1% (no primary revascularization) and 27.9% (surgical revascularization) (Appendix A; primary endpoint by primary revascularization and by primary lesion at limb level in Appendix A).

The occurrence of minor limb events was 12.2% overall, with a median time to minor limb events in cases of occurrence of 29.5 days (IQR 15–79.5). The occurrence of a minor limb event was most frequent in cases of isolated infrapopliteal primary lesions (17.1%) (Table 3, Figure 2B; secondary endpoint by primary revascularization and by primary lesion at limb level in Appendix A).

In the multivariable Cox regression, no adjustment for age, sex, and diabetes mellitus was included in the model for the major adverse limb event occurrence due to non-inclusion in the backfitting algorithm. The risk of a major adverse limb event increased with frailty (frailty 3 HR 1.14, 95% CI 1.02–1.28). In reference to the primary lesion being found in all three locations, the risk of a major adverse limb event was lower in cases of infrainguinal-to-popliteal lesions (HR 0.51, 95% CI 0.27–0.98) and was equally likely in all other cases. In reference to no primary revascularization, the risk of a major adverse limb event was significantly increased in all types of primary revascularization procedures (highest HR for interventional revascularizations 2.16, 95% CI 1.70–2.74) (Figure 1B and Figure 3B, Appendix A).

The risk of a minor limb event increased in cases of the presence of diabetes mellitus (HR 1.49, 95% CI 1.01–2.21) and with frailty (frailty 3 HR 5.15, 95% CI 3.00–10.09) and was independent of age in the multivariable Cox regression, in which an adjustment for sex was eliminated. In the same model, the risk of minor limb events increased after a surgical (HR 1.14, 95% CI 1.06–1.22) or both interventional and surgical revascularization (HR 1.14, 95% CI 1.04–1.24). The risk of minor limb events was highest in cases of isolated primary infrapopliteal lesions and was less likely in cases of all other primary types of lesions (Figure 2B, Appendix A).

## 4. Discussion

This high-volume single-center study of 2067 patients suffering from PAD investigated the association between patterns of atherosclerotic lesions in lower extremities and outcome parameters at both patient and limb levels in an median observation period of 1020 days. At the patient level, the lowest overall survival and amputation-free survival was found in cases of abundant lower extremity atherosclerotic lesions or in cases of isolated infrapopliteal atherosclerotic lesions. The latter subgroup accounted for 13.3% of all patients; this PAD subgroup might profit from intensified measures of cardiovascular risk reduction. This finding, however, warrants further prospective studies due to limitations within the study design of this analysis. In addition, the same subgroup was at the highest risk of minor adverse limb events. The abundancy of atherosclerotic lesions, on the other hand, was associated with the highest risk of a major adverse limb events, and this risk was lowest in cases of isolated infrainguinal-to-popliteal lesions.

PAD affects the arteries distal to the aortic bifurcation with inter-individual heterogeneity of atherosclerotic lesions, which is in parts explained by factors such as individual anatomic/hemodynamic, cellular, and/or biochemical processes [27,28]. For example, suprainguinal arteries are more elastic than more distal arteries, while infrapopliteal arteries contain more muscular elements. As a result, an association between risk factors and different types of vessel diameters has been found [29]. Among other observations, and with some inconsistency, it has been noted that atherosclerotic lesions in large vessels occur at a younger age and may be more progressive and associated with inflammatory markers, whereas smoking and hypercholesterolemia are associated with large-vessel disease. Diabetes mellitus and renal insufficiency, on the other hand, are associated with small-vessel disease, which also occurs with increasing age, while hypertension may be associated with distal lesions in cerebral arteries [11,14,27,30,31,32,33,34,35,36,37,38]. In the current study, chronic kidney disease and diabetes mellitus were most frequent in isolated infrapopliteal lesions, hypercholesterolemia was most frequent in infrainguinal-to-popliteal lesions and lesions in all three locations, and patients with isolated infrapopliteal lesions in either of the two lower extremity limbs were older than other subcohorts of allocated primary lesions, which is in line with the existing literature. In addition, in the current analysis, the peripheral atherosclerotic burden was associated with the comorbidity of coronary vessels; in cases of lesions at all levels, the fraction of comorbid coronary heart disease was 46.8%, while it was 20.3% in cases of isolated suprainguinal lesions. While these findings are based on a retrospective analysis, the clinical implications of these findings could highlight the need for risk stratification by the lesion location and burden in PAD patients. Even though all PAD patients are at risk of cardiovascular death, patients with a more advanced disease, i.e., lesions in all locations of at least one lower extremity, and patients with an isolated infrapopliteal lesion alike might benefit from an extensive cardiovascular risk reduction.

The evidence of an association between lesions or lesion patterns and outcome parameters remains partly contradicting; while some studies have found poor prognoses, especially in cases of distal disease [38], other studies have found evidence of mortality rates for proximal lesions being up to two to seven times higher than for distal lesions [39], or no impact of the disease location on survival was found [15,40]. Independently of the lesion location in the lower extremities, multivascular atherosclerotic disease (cerebral, coronary arterial), which is frequent in symptomatic PAD patients, was associated with an increased risk of all-cause and cardiovascular mortality [41], and PAD patients have been found to have a high risk of bilateral disease [42]. In terms of stratified risk in PAD at the patient and limb level, in a Japanese cohort, no trend in overall survival was found in a PAD cohort by the location of the atherosclerotic lesion, while major adverse limb events depended on the number of treated locations [40]. Ozkan et al. were able to demonstrate that distal lesions in particular are closely associated with the development of chronic limb-threatening ischemia and found that major adverse limb events were more likely to occur [37]. In a Dutch cohort, a number of lesions higher than or equal to three lesions was associated with an increased risk of a composite endpoint of cardiovascular events over three years of follow-up [16].

In the current analysis, the overall risk of death was highest in cases of isolated infrapopliteal lesions, lesions in two of three locations, or lesions in all three locations and is therefore in line with the major body of existing evidence. To analyze differences among study results, though, different aspects must be considered.

First, the definition of the level allocation must be taken into account. Different studies have defined atherosclerotic lesions of any kind to be lesions of interest or have focused on lesions leading to intervention, while in the current study, atherosclerotic lesions were only deemed relevant if they were explicitly described as stenoses or if they were revascularizable/revascularized. It was not the intention of this analysis to analyze the impact of the extent or length of lesions on outcome parameters [8,43]; we explicitly intended to investigate the association between lesion location patterns, which are easily reportable and interpretable among healthcare professionals not specialized in vascular medicine.

Second, among studies, level allocation has been conducted using different diagnostic modalities, such as the ankle–brachial index, ultrasound results, or, like was the case in this analysis, angiograms. In this context, angiograms are considered the gold standard in PAD diagnostics.

Third, outcome parameters differ among existing studies. Amputation-free survival was chosen as the primary endpoint at the patient level, introducing a possible bias for a subset of patients initially treated at the University Hospital Leipzig and sequentially receiving amputation surgery at a different center, and overall survival was defined to be the secondary endpoint at the patient level. Since survival data were obtained from the official registration office, this endpoint was not biased by re-admissions to different hospitals.

Beyond the lesion location, in the current analysis, diabetes mellitus and male sex were independently associated with an increased risk of death; in addition, in the presence of diabetes mellitus, the risk of minor limb events was increased [44,45]. Also, notably, primary isolated interventional revascularization was associated with a reduced risk of death at the patient level, while an increased risk of a major adverse limb event was found at the limb level following any revascularization. The increased risk of death (except for interventional treatment) or amputation found in patients following any invasive treatment must be interpreted as more advanced atherosclerotic lesions in these limbs and/or patients. Besides patient symptoms, age, and the extent of comorbidities, the atherosclerotic lesion location is decisive in the selection of invasive therapeutic approaches [8,9,10,11], i.e., interventional vs. surgical vs. combined techniques, while medication and conservative risk reduction are, to date, independent of the lesion location, even though the pathophysiology and preventive effects of established therapies may be dependent on atherosclerotic lesion patterns [12,13], especially in PAD vs. coronary atherosclerosis [46,47]. The addition of medication (anticoagulation, statin) to the multivariable models at the patient level in subgroups with information on these parameters did not increase the model accuracy in a likelihood ratio test in this analysis, while other covariates were minimally changed after the inclusion of these parameters. However, in 60% of patients, this information was lacking. The same was the case for chronic kidney disease and body weight in this analysis.

The clinical presentation of claudication vs. CLTI is known to be decisive for the risk of mortality and amputation and is related to PAD progression. In CLTI, in the current analysis, the risk of both major amputation and death was increased in comparison to claudicants. There was an association between this stratification and the lesion allocation: in cases of isolated infrapopliteal lesions or lesions at all levels, CLTI (surrogated by Fontaine III or IV) was present in 72.4% and 55.4%, respectively, while it was present in 15.1% in cases of isolated suprainguinal lesions. This is in line with the existing literature; for instance, current ESC guidelines address this association and offer recommendations stratified by the lesion location and lesion complexity [48]. Claudication, on the other hand, was most common (85.0%) in cases of isolated suprainguinal lesions, while it was least frequent (27.5%) in infrapopliteal lesions. However, the current analysis demonstrated the impact of lesion allocation on mortality and amputation after the adjustment for this clinical stratification of CLTI vs. claudication.

### 4.1. Limitations

Beyond limitations stated above, this study has other relevant limitations to consider. First, there was no side coding of PAD stages in the ICD diagnoses, making it impossible to obtain PAD Fontaine stage information at the limb level. In 2020, the SARS-CoV-2 virus and the resulting COVID-19 disease led to a worldwide pandemic, affecting Germany until late 2021, which was partly during the observation period of this analysis. During this time, numerous chronic conditions were suboptimally treated, including PAD patients [49], while overall time trends in PAD treatment were maintained during the COVID-19 pandemic in Germany [50]. Also, no laboratory results could be accessed due to a lack of availability, information on medication was only available for 40% of the cohort, and no information on smoking history was available. The latter especially is likely to introduce residual confounders to the results of our analysis [29,51]. In addition, functional clinical data, i.e., duplexsonography results, the ankle brachial index, pain scores, and walking distances, were unavailable. Also, collateral vessel development could not be analyzed. The universality of the results of this study may be restricted due to various factors, including different therapeutic strategies among countries in cases of hyperlipidemia, fractions of smokers, and strategies for and the availability of revascularization.

### 4.2. Conclusions

In this retrospective single-center analysis from a high-volume German center, an in-depth analysis of lower extremity PAD’s long-term outcomes was performed both at the limb level and at the patient level. Lower extremity atherosclerotic lesion patterns were found to be associated with distinct comorbidity structures and differing amputation-free and overall survivals at the patient level, which were lowest in cases of infrapopliteal lesions, two of three, or all three affected levels in lower extremity PAD. A different pattern was found at the limb level with regard to major adverse limb events; the highest risk was found in cases of atherosclerotic lesions in all three locations, and a reduced risk was only observed in cases of isolated infrainguinal-to-popliteal lesions. The highest risk of minor limb events was identified in isolated infrapopliteal lesions. While the study design precludes us from making definitive recommendations, these findings warrant further prospective studies to elaborate on the clinical implications of this risk stratification.

## Figures and Tables

**Figure 1 jcm-14-07037-f001:**
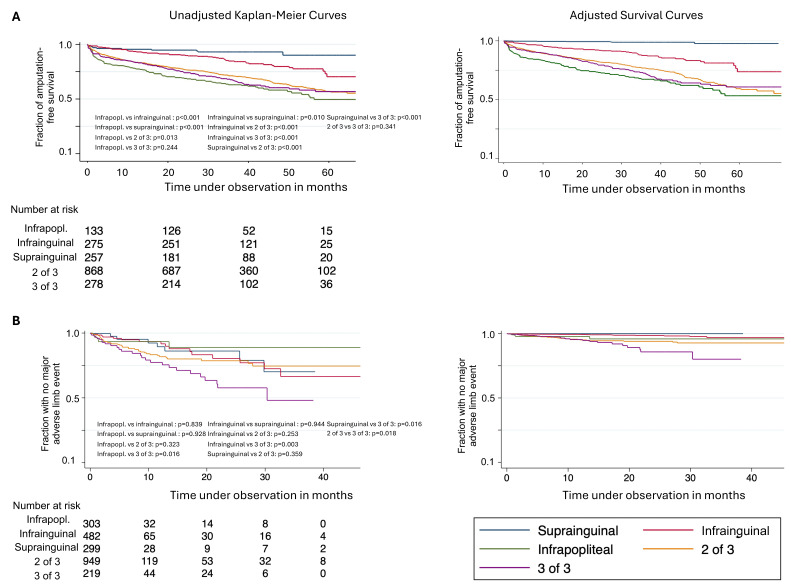
Primary endpoints. (**A**) Primary endpoint at patient level: Amputation-free survival by primary anatomical location of arterial atherosclerotic lesion. (**B**) Primary endpoint at limb level: Time without major adverse limb events by primary anatomical location of arterial atherosclerotic lesion. Kaplan–Meier Curves and adjusted survival curves are depicted. Infrapopl. for infrapopliteal lesion. Infrainguinal short for infrainguinal-to-popliteal; 2 of 3 for lesion at two levels, 3 of 3 for lesion at all levels. *p* values from log-rank tests.

**Figure 2 jcm-14-07037-f002:**
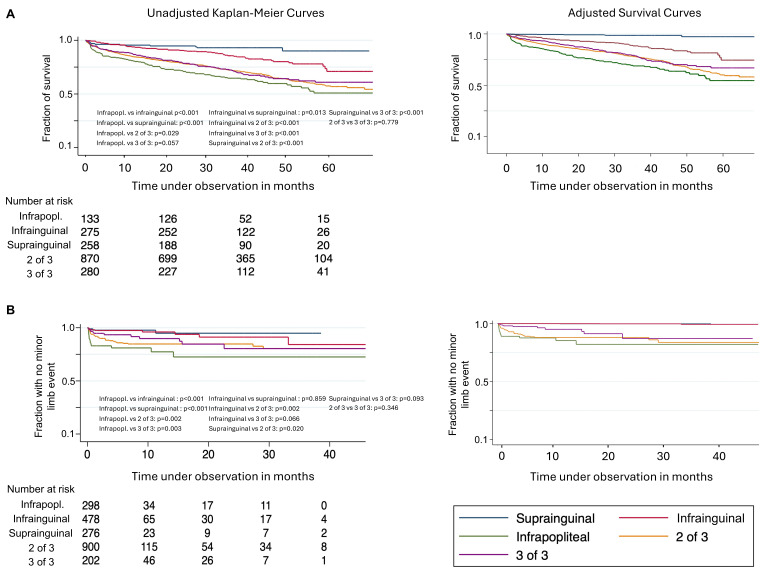
Secondary endpoints. Secondary endpoint at patient level: overall survival in (**A**); minor limb events as secondary endpoint at limb level in (**B**) by primary anatomical location of arterial atherosclerotic lesion. Kaplan–Meier Curves and adjusted survival curves are depicted. Infrapopl. for infrapopliteal lesion. Infrainguinal short for infrainguinal-to-popliteal; 2 of 3 for lesion at two levels and 3 of 3 for lesion at all levels. *p* values from log-rank tests.

**Figure 3 jcm-14-07037-f003:**
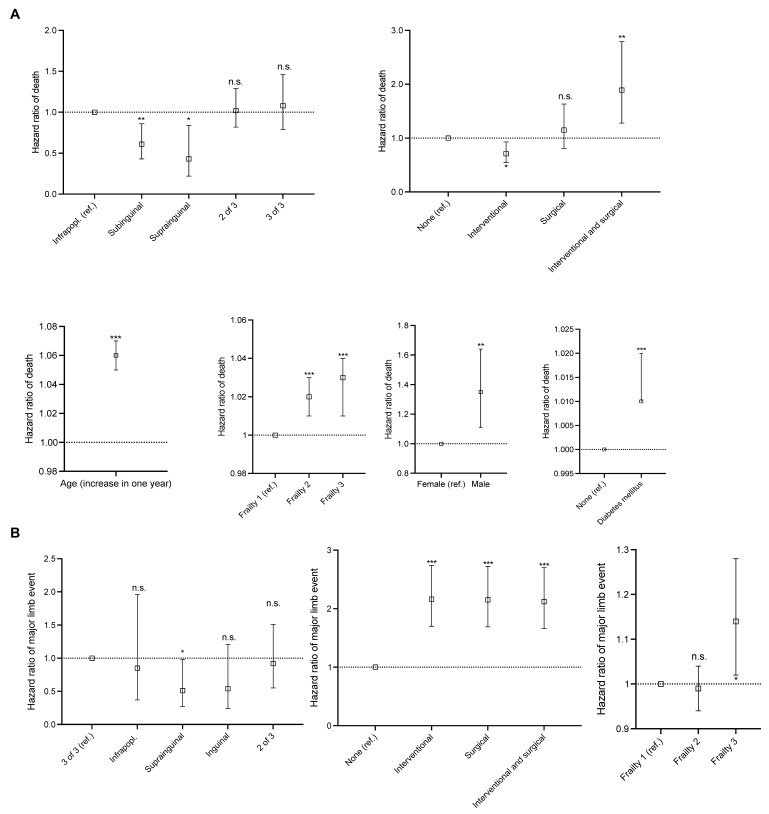
Results from a multivariable Cox regression: primary endpoints. (**A**) Primary endpoint at patient level: amputation-free survival. (**B**) Primary endpoint at limb level: occurrence of major adverse limb event. Univariable and multivariable Hazard Ratios for (**A**) in Appendix A and univariable and multivariable Hazard Ratios for (**B**) in Appendix A. Secondary endpoints at patient and limb level in Appendix A, corresponding univariable and multivariable Hazard Ratios in Appendix A and Appendix A, respectively. Ref. for reference. Infrapopl. for infrapopliteal. n.s. for not significant. The dashed lines represent Hazard Ratio = 1. * for *p* < 0.05, ** for *p* < 0.01, and *** for *p* < 0.001.

**Table 1 jcm-14-07037-t001:** Overall cohort: patient characteristics. Values in parentheses are percentages of total in the patient group unless otherwise indicated; * values are median (interquartile range). ^†^ Available data for medication was n = 841 and for BMI n = 286. In case of medication, the total (100%) is considered 841.

Total no. of patients	2067
Age at first admission n (%), overall *	68 (61–78)
≤59	404 (19.6)
60–74	961 (46.5)
≥75	702 (34.0)
No. of females n (%)	595 (28.8)
Sex ratio (M/F)	2.5:1
Body mass index kg/m^2^ *	26 (23–30)
PAD stage at first admission (Fontaine) n (%)	
IIa	94 (4.6)
IIb	926 (44.9)
III	267 (13.0)
IV	732 (35.5)
unknown	48 (2.3)
Acute limb ischemia n (%)	37 (1.8)
No of admissions per patient n (%), overall *	1 (1–1)
1	1411 (68.3)
2	400 (19.4)
≥3	258 (12.5)
Time between admissions *	63 (34–156)
Frailty at first admission n (%)	
1	1518 (73.4)
2	433 (21.0)
3	116 (5.6)
Coronary heart disease n (%)	753 (36.4)
Diabetes mellitus n (%)	941 (45.5)
Arterial hypertension n (%)	1610 (77.9)
Hypercholesterolemia n (%)	1555 (75.2)
Chronic kidney disease n (%)	951 (46.0)
Medication ^†^	
Antiplatelet any n (%)	459 (54.6)
Antiplatelet mono n (%)	313 (37.2)
DOAK or Vit.K-antagonist n (%)	71 (8.4)
Statin n (%)	595 (70.8)

**Table 2 jcm-14-07037-t002:** Patient level: Primary and secondary endpoints and procedures performed by primary anatomical location of arterial atherosclerotic lesion. All numbers are n (%) and refer to total of columns except in the upper row, where they are n (%) of the row total. * values are median (interquartile range). The stated level of lesions refers to either side of the lower limbs. Since for some revascularization procedures there was no side coding, primary revascularizations by primary anatomical location of arterial atherosclerotic lesion do not add up to the total of the respective revascularizations. Time under observation refers to patient status of the primary endpoint, i.e., time under observation or time to death or time to amputation above the ankle, stratified by primary revascularization in Appendix A. Numbers smaller than 3 were changed into a range. The percentage was randomly chosen as that of one of the numbers within the range.

	Overall Cohort	Missing Allocation	Suprainguinal Lesion	Infrainguinal-to-Popliteal Lesion	Infrapopliteal Lesion	Lesion At Two Levels	Lesion at All Levels
No. of patients	2067	251 (12.1)	133 (6.4)	275 (13.3)	258 (12.5)	870 (42.1)	280 (13.6)
Primary revascularization							
None	1502 (72.7)	94 (37.5)	124 (93.2)	224 (81.5)	233 (90.3)	652 (74.9)	175 (62.5)
Interventional	253 (12.2)	13 (5.2)	6 (4.5)	35 (12.7)	24–26 (9.7)	133 (15.3)	42 (15.0)
Surgical	215 (10.4)	108 (43.0)	0–2 (0.8)	14 (5.1)	0–2 (0.4)	52 (6.0)	39 (13.9)
Both interventional + surgical	97 (4.7)	36 (14.3)	1–3 (1.5)	1–3 (0.7)	0	33 (3.8)	24 (8.6)
Time under observation (in days; min–max, median, IQR)	0–2149, 1020, 658–1412	0–2147, 139, 16–1047	2–2099, 1092, 879–1312	3–2148, 1168, 895–1540	3–2125, 951, 457–1311	0–2147, 1073, 736–1477	1–2149, 1040.5, 693–1411
In-hospital death n (%)	97 (4.7)	35 (13.9)	3–5 (2.3)	2–4 (0.7)	15 (5.8)	34 (3.9)	8 (2.9)
Endpoints at patient level:							
Amputation-free survival	1398 (67.6)	150 (59.8)	123 (92.5)	224 (81.5)	153 (59.3)	572 (65.7)	176 (62.9)
Time to major amputation or death (days) *	340, 58–843	52, 7–497	162, 29–814	610, 250–1052	289.5, 42.5–678	427, 92–927	445.5, 88–828
Overall survival	1434 (69.4)	160 (63.8)	123 (92.5)	225 (81.8)	158 (61.2)	579 (66.6)	189 (67.5)
Time to death (days) *	407, 88–899	127 (16–596)	162 (34–814)	652.5 (293–1096)	358 (67.5–731)	445 (133–961)	483 (113–930)

**Table 3 jcm-14-07037-t003:** Limb level: Primary and secondary endpoints and procedures performed by primary anatomical location of arterial atherosclerotic lesion. All numbers are n (%) and refer to total of columns except in the upper row, where they are n (%) of the row total. * values are median (interquartile range). Since for some revascularization procedures there was no side coding, primary revascularizations by primary anatomical location of arterial atherosclerotic lesion do not add up to the total of the respective revascularizations. Time under observation refers to limb status of the primary endpoint, i.e., time under observation or time to major adverse limb event, stratified by primary revascularization in Appendix A. Numbers smaller than 3 were changed into a range. The percentage was randomly chosen as that of one of the numbers within the range.

	Overall Cohort	Missing Allocation	Suprainguinal Lesion	Infrainguinal-to-Popliteal Lesion	Infrapopliteal Lesion	Lesion at Two Levels	Lesion at All Levels
No. of limbs	2633	380 (14.4)	304 (11.6)	482 (18.3)	299 (11.4)	949 (36.0)	219 (8.3)
Primary revascularization							
None	2189 (83.1)	134 (35.3)	290 (95.4)	450 (93.4)	291 (97.3)	851 (89.7)	173 (79.0)
Interventional	127 (4.8)	66 (17.4)	8 (2.6)	16 (3.3)	7–9 (2.7)	22 (2.3)	8 (3.7)
Surgical	233 (8.9)	137 (36.1)	6 (2.0)	12 (2.5)	1–3 (0.3)	50 (5.3)	27 (12.3)
Both interventional + surgical	84 (3.2)	43 (11.3)	0	4 (0.8)	0	26 (2.7)	11 (5.0)
Time under observation for limb events (in days; min–max, median, IQR)	0–1423, 12, 4–78	0–1257, 32, 8.5–147	0–1157, 4, 2–30.5	1–1413, 6, 2–71	1–1413, 9, 4–46	1–1423, 13, 4–77	1–1148, 34, 6–170
Endpoints at limb level:							
Major adverse limb event	164 (6.2)	55 (14.5)	8 (2.6)	17 (3.5)	8 (2.7)	48 (5.1)	28 (12.8)
Time to major adverse limb event (days) *	122.5 (39.5–354.5)	81 (21–352)	321.5 (120.5–576.5)	364 (65–523)	29.5 (18.5–41.5)	89 (50–248)	188.5 (54.5–425)
Minor limb event	320 (12.2)	111 (29.2)	9 (3.0)	13 (2.7)	51 (17.1)	106 (11.2)	30 (13.7)
Time to minor limb event (days) *	29.5 (15–79.5)	0 (0–10)	0 (0–4)	12 (0–272)	4 (0–8)	4.5 (0–19)	0 (0–27)

## Data Availability

The datasets presented in this article are not readily available because, despite the anonymization process, re-identification of individual patients may be possible based on patient characteristics. The analysis script (Stata Version 16) can be made accessible after direct contact with Konstantin Uttinger at konstantin.uttinger@unimedizin-ffm.de and after an evaluation of the request.

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
