# Peer review of "Outcomes at Patient and Limb Levels in Peripheral Artery Disease by the Location of Atherosclerotic Lower Limb Lesions: An Observational Study from a High-Volume German Center"

_jcm, 2025, doi:10.3390/jcm14197037_

Round 1

Reviewer 1 Report

Comments and Suggestions for Authors

I reviewed with interest the manuscript by Anne Zimmermann et al. "Outcome at patient- and limb-level in Peripheral Artery Disease by location of atherosclerotic lower limb lesions: An observational study from a German high-volume center". In this article, the authors address an old topic - they study the prognostic value of the lesion level in PAD. The authors obtained the expected results - peripheral and multilevel lesions in these patients are associated with the worst prognosis, both at the patient and limb levels. A distinctive feature of this study is the large cohort of patients examined, as well as the use of angiography as a method for assessing the presence of stenosis in PAD. This makes the conclusions obtained by the authors sufficiently evidentiary.

While reviewing, I had the following comments and questions:

  1. In the Introduction section, the novelty of the study is not sufficiently presented. The authors do not provide conflicting results of previous studies that would justify the need for this study. There may have been some additional considerations for choosing this study design, which should have also been written about in this section.
  2. The authors provide extensive tabular material, which can only be welcomed. However, the format of data presentation in Table 2 makes it difficult to perceive. The headings in the column names need to be corrected. I also do not understand the icons used by the authors in the table - for example, the arrows in different directions.
  3. The text of the Results section is overloaded with numerical data, which makes it difficult to perceive when reading. It is sufficient to provide links to tables with minimal numbers to illustrate the results under consideration.
  4. The authors widely cite studies from years ago, but do not pay due attention to recent publications on the topic. In my opinion, recent publications (for example, ref. 1-2, see below) deserve consideration in the Discussion section.

5.. Subheadings for the Limitations of the Study and Conclusion sections should be added to the manuscript.

  1. Smolderen KG, van Zitteren M, Jones PG, Spertus JA, Heyligers JM, Nooren MJ, Vriens PW, Denollet J. Long-Term Prognostic Risk in Lower Extremity Peripheral Arterial Disease as a Function of the Number of Peripheral Arterial Lesions. J Am Heart Assoc. 2015 Oct 26;4(10):e001823. doi: 10.1161/JAHA.115.001823.
  2. Niiranen O, Virtanen J, Rantasalo V, Ibrahim A, Venermo M, Hakovirta H. The Association between Major Adverse Cardiovascular Events and Peripheral Artery Disease Burden. J Cardiovascular Dev Dis. 2024 May 21;11(6):157. doi: 10.3390/jcdd11060157.

Reviewer 2 Report

Comments and Suggestions for Authors

The manuscript presents a large single-center retrospective cohort study of patients undergoing endovascular revascularization for peripheral arterial disease (PAD). The overall novelty of the work is moderate, as most findings confirm previously established associations between infrainguinal lesion outcomes, limb-level risk, and Fontaine stage. Its primary contribution lies in providing detailed limb-level outcomes in a sizeable cohort, which adds some value to the existing literature.

However, the study is limited by the absence of several important clinical measures that strongly influence outcomes and interpretation. Specifically, there are no data on pharmacological therapy (antiplatelets, statins, anticoagulants), biochemical markers (HbA1c, creatinine, lipid profile), or lifestyle factors such as smoking. Functional assessment is also restricted to Fontaine stage, without complementary measures such as ankle–brachial index, walking distance, or pain scores.

The statistical methods are generally appropriate, with Cox models and adjustments suitably applied. Nonetheless, the omission of key covariates, particularly smoking status and renal function, raises concerns about residual confounding.

Overall, the conclusions are broadly consistent with the presented data, but their interpretation should be tempered. Greater emphasis is needed on the limitations of the retrospective design, the absence of critical clinical and laboratory data, and the restricted generalizability of the findings. Claims regarding clinical applicability or risk stratification should therefore be moderated accordingly.

Round 2

Reviewer 1 Report

Comments and Suggestions for Authors

The authors responded to my questions and comments and made corrections to the manuscript. I have no other comments.

Reviewer 2 Report

Comments and Suggestions for Authors

The authors have addressed the raised concerns to the best of their ability and revised the manuscript accordingly. I consider it suitable for publication in its current form.